

# ensembleTax: an R package for determinations of ensemble taxonomic assignments of phylogenetically-informative marker gene sequences

Dylan Catlett   Kevin Son   Connie Liang

Earth Research Institute, University of California, Santa Barbara, Santa Barbara, CA, United States of America

Corresponding author
Dylan Catlett, dsc@ucsb.edu

## ABSTRACT

**Background**. High-throughput sequencing of phylogenetically informative marker genes is a widely used method to assess the diversity and composition of microbial communities. Taxonomic assignment of sampled marker gene sequences (referred to as amplicon sequence variants, or ASVs) imparts ecological significance to these genetic data. To assign taxonomy to an ASV, a taxonomic assignment algorithm compares the ASV to a collection of reference sequences (a reference database) with known taxonomic affiliations. However, many taxonomic assignment algorithms and reference databases are available, and the optimal algorithm and database for a particular scientific question is often unclear. Here, we present the ensembleTax R package, which provides an efficient framework for integrating taxonomic assignments predicted with any number of taxonomic assignment algorithms and reference databases to determine ensemble taxonomic assignments for ASVs.

**Methods**. The ensembleTax R package relies on two core algorithms: *taxmapper* and *assign.ensembleTax*. The *taxmapper* algorithm maps taxonomic assignments derived from one reference database onto the taxonomic nomenclature (a set of taxonomic naming and ranking conventions) of another reference database. The *assign.ensembleTax* algorithm computes ensemble taxonomic assignments for each ASV in a data set based on any number of taxonomic assignments determined with independent methods. Various parameters allow analysts to prioritize obtaining either more ASVs with more predicted clade names or more robust clade name predictions supported by multiple independent methods in ensemble taxonomic assignments.

**Results**. The ensembleTax R package is used to compute two sets of ensemble taxonomic assignments for a collection of protistan ASVs sampled from the coastal ocean. Comparisons of taxonomic assignments predicted by individual methods with those predicted by ensemble methods show that conservative implementations of the ensembleTax package minimize disagreements between taxonomic assignments predicted by individual and ensemble methods, but result in ASVs with fewer ranks assigned taxonomy. Less conservative implementations of the ensembleTax package result in an increased fraction of ASVs classified at all taxonomic ranks, but increase the number of ASVs for which ensemble assignments disagree with those predicted by individual methods.

**Discussion**. We discuss how implementation of the ensembleTax R package may be optimized to address specific scientific objectives based on the results of the application

of the ensembleTax package to marine protist communities. While further work is required to evaluate the accuracy of ensemble taxonomic assignments relative to taxonomic assignments predicted by individual methods, we also discuss scenarios where ensemble methods are expected to improve the accuracy of taxonomy prediction for ASVs.

## INTRODUCTION

High-throughput amplicon sequencing of phylogenetically informative marker genes (also known as DNA meta-barcoding) is a widely used method for assessing the composition and diversity of microbial communities (*Sogin et al., 2006*; *De Vargas et al., 2015*). Commonly used phylogenetic marker genes include the 18S small subunit ribosomal RNA gene (18S rDNA) in microbial eukaryotes, the 16S small subunit ribosomal RNA gene (16S rDNA) in prokaryotes, and the internal transcribed spacer region (ITS) in fungi (*Woese & Fox, 1977*; *Medlin et al., 1988*; *Sogin et al., 2006*; *De Vargas et al., 2015*). Marker gene sequencing studies typically rely on operational taxonomic units or amplicon sequence variants (ASVs) to serve as representatives of individual microbial species within a community (*Schloss et al., 2009*; *Callahan et al., 2016*); hereafter we use "ASV" to denote a marker gene sequence sampled from any system). Because an organism's taxonomy is often correlated with its ecology, assigning taxonomic identities to ASV sequences imparts ecological significance to genetic data. Taxonomic assignment thus represents a critical component of all marker gene sequencing studies.

Taxonomic assignment of marker gene sequences requires a reference database of marker gene sequences with known taxonomic identities, and an assignment algorithm that determines the most likely taxonomic affiliation of each representative ASV in a data set by comparing it to the sequences in a reference database. Reference databases are typically tailored to a specific group of organisms and/or a single marker gene (*Guillou et al., 2013*; *Quast et al., 2013*; *Cole et al., 2014*; *Glöckner et al., 2017*), although some include multiple marker genes from multiple groups of organisms (e.g., the SILVA reference database; *Quast et al., 2013*; *Glöckner et al., 2017*). However, different reference databases frequently employ disparate taxonomic naming and ranking conventions, and certain reference databases include subsets of reference sequences that are not found in other databases. For example, at the time of writing both the Protistan Ribosomal Reference database (pr2; *Guillou et al., 2013*) and the SILVA reference database (silva; *Quast et al., 2013*; *Glöckner et al., 2017*) include large collections of 18S rDNA reference sequences. The disparate naming and ranking conventions employed by silva and pr2 make it difficult for analysts of 18S rDNA data sets to reconcile taxonomic assignments predicted using one database with those predicted using the other.

Analysts of marker gene sequence data sets must also choose from one of many taxonomic assignment algorithms, each of which employs unique methods and is associated with error (e.g., *Wang et al., 2007*; *Bokulich et al., 2018*; *Edgar, 2018a*; *Murali, Bhargava & Wright, 2018*). There are some assignment algorithms that consistently outperform others across multiple benchmark exercises, such as the RDP naïve Bayesian classifier (*Wang et al., 2007*), the recently introduced idtaxa algorithm (*Murali, Bhargava & Wright, 2018*), and others. However, realistic validation of taxonomic assignment algorithms is not straightforward (see *Edgar, 2018a* for a summary of approaches used in benchmarking assignment algorithms). Thus, the optimal taxonomic assignment algorithm and reference database for a particular scientific question or data set is often uncertain.

Ensemble approaches that integrate results from multiple independent bioinformatic methods have been shown to improve the accuracy of assigning taxonomy to meta-genomic fragments, and of assigning meta-genomic fragments to genomes (*McIntyre et al., 2017*; *Sieber et al., 2018*). Similarly, ensemble approaches that incorporate taxonomic assignments from multiple taxonomic assignment algorithms and/or reference databases may be expected to improve taxonomic assignments of ASV sequences. To our knowledge, only one ensemble (or "consensus") approach has been proposed for taxonomic assignment of fungal ITS sequences (*Gdanetz et al., 2017*), but this method is not generalizable to prokaryotic 16S rDNA or eukaryotic 18S rDNA sequences and does not enable the use of multiple reference databases with disparate naming and ranking conventions.

Here we introduce the ensembleTax R package, which enables analysts of marker gene sequence data to efficiently and flexibly compute ensemble (or "consensus") taxonomic assignments for each representative ASV sequence in a marker gene data set. Two core algorithms, *taxmapper* and *assign.ensembleTax*, allow users to map taxonomic assignments from one reference database onto another reference database's taxonomic nomenclature (see Table 1 for a glossary of terms used here), and to compute ensemble taxonomic assignments, respectively. The ensembleTax R package includes *taxmapper*, *assign.ensembleTax*, and additional pre-processing functions that enable streamlined ensemble taxonomic assignment determinations immediately following determination and initial taxonomic assignment of ASVs with the dada2 and DECIPHER R packages (*Callahan et al., 2016*; *Wright, 2016*; *Murali, Bhargava & Wright, 2018*). We demonstrate the utility and flexibility of the ensembleTax R package using a collection of protistan ASVs derived from the V9 hypervariable region of the 18S rDNA sampled from the coastal ocean. In particular, we demonstrate how the ensembleTax R package allows investigators to prioritize obtaining either more predicted clade names for more ASVs or more robust taxonomic assignment predictions supported by multiple independent methods in ensemble taxonomic assignments. The ensembleTax R package is freely available on GitHub (https://github.com/dcat4/ensembleTax/blob/master/README.md), and CRAN (https://cran.rstudio.com/web/packages/ensembleTax/index.html), and will continue to be developed as taxonomic assignment methods and reference databases continue to evolve.
**Table 1  Definitions of terms used in the present manuscript and the ensembleTax package.**

| Term | Definition |
|------|-----------|
| Taxonomic nomenclature | A specific framework of naming and ranking conventions employed by a reference database. |
| Taxonomy table | A collection of ASV sequences (or ASV identifiers) sampled from the environment with corresponding taxonomic assignments. |
| Taxonomy | An organism's taxonomic identity. |
| Taxonomic assignment | The predicted taxonomy for an ASV. |
| Ensemble taxonomic assignment | A taxonomic assignment for an ASV determined from several independent taxonomic assignment methods. Synonymous with "consensus taxonomic assignment". |
| Taxonomic synonyms | Taxonomic names with equivalent meaning. |
| Lower/higher taxonomic ranks | "Lower" taxonomic ranks would correspond to lower branches in a phylogenetic tree (e.g., species is a lower rank than class). |

# PACKAGE DESCRIPTION AND IMPLEMENTATION

## ensembleTax package overview

The ensembleTax R package is implemented in the R computing language and was developed and built with R version 3.6.2 (*R Core Team, 2019*). The R package devtools (*Wickham, Hester & Chang, 2020*) was used extensively to build the ensembleTax package. The ensembleTax package relies on other R packages including dplyr (*Wickham et al. 2020*), stringr (*Wickham, 2019*), DECIPHER (*Wright, 2016*), Biostrings (*Pagès et al., 2019*), and usethis (*Wickham & Bryan, 2020*). Vignettes were built with the knitr (*Xie, 2020*) and rmarkdown (*Allaire et al., 2020*) packages. The ensembleTax package is freely available for use under an MIT license, and can be installed from GitHub or CRAN (see https://github.com/dcat4/ensembleTax/blob/master/README.md for download instructions).

The ensembleTax R package was developed to offer an automated, efficient, and flexible tool for microbial ecologists to synthesize taxonomic assignment predictions made with any number of taxonomic assignment algorithms and/or reference databases. Table 1 provides a glossary of terms used here and within the documentation of the ensembleTax package, and Fig. 1 outlines possible ensembleTax workflows for different taxonomic assignment methods. All ensembleTax workflows consist of converting the outputs of taxonomic assignment algorithms into dataframes that include ASV-identifying meta-data and corresponding taxonomic assignments for an arbitrary number of ranks, mapping of taxonomic assignments onto a common taxonomic nomenclature (if necessary), and determination of ensemble taxonomic assignments (Fig. 1). The ensembleTax package was developed to offer convenient analysis following ASV determination and initial taxonomic assignment with the dada2 and DECIPHER R packages (*Callahan et al., 2016*; *Wright, 2016*; *Murali, Bhargava & Wright, 2018*), but other methods for ASV determination and initial taxonomic assignment may be used prior to the ensembleTax workflow (Fig. 1).
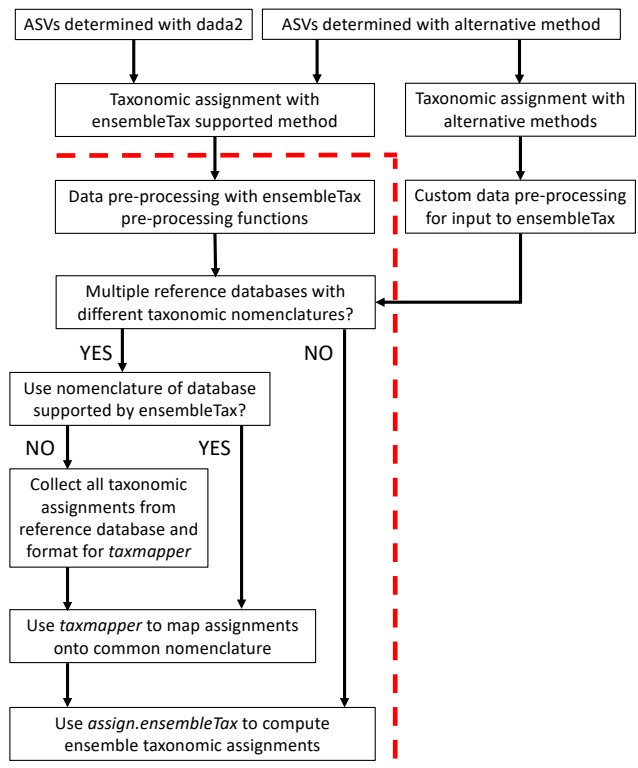

**Figure 1** **Schematic diagram of potential ensembleTax R package workflows.** The red dashed line separates components of the amplicon sequence data analysis workflow that must be completed prior to use of the ensembleTax package from those that can be addressed with the ensembleTax package.

The taxonomic assignment algorithms explicitly supported in the ensembleTax package are the Ribosomal Database Project (RDP) naïve Bayesian classifier *Wang et al., 2007* implemented in the dada2 R package (*Callahan et al., 2016*), and the recently developed idtaxa algorithm (*Murali, Bhargava & Wright, 2018*) implemented in the DECIPHER R package (*Wright, 2016*). Explicitly supported reference databases include the SILVA SSU non-redundant reference database v138 (*Quast et al., 2013*; *Glöckner et al., 2017*; henceforth, silva), the Protistan Ribosomal Reference database v4.12.0 (*Guillou et al., 2013*; henceforth, pr2), the GreenGenes reference database v13.8 clustered at 97% similarity (*DeSantis et al., 2006*; *McDonald et al., 2012*), and the RDP 16S rRNA training set v16 (*Cole et al., 2014*). Additional assignment algorithms and/or reference databases can be incorporated so long as the data can be read into R and converted to a dataframe object. We welcome external contributions of R implementations of other taxonomic assignment algorithms and/or reference database data, and will continue to accommodate new methods as they are developed.

## Description of core algorithms and package data

As noted above, *taxmapper* and *assign.ensembleTax* are the core algorithms for computing ensemble taxonomic assignments with the ensembleTax package. *taxmapper* maps, or

"translates", the taxonomic nomenclature used by one reference database onto the taxonomic nomenclature used by another, while *assign.ensembleTax* computes ensemble taxonomic assignments from taxonomic assignments generated by any number of unique taxonomic assignment algorithm and reference database combinations. Here we describe these algorithms and parameters that may be altered to tailor their performance for particular scientific objectives.

*taxmapper* maps the taxonomic nomenclature of one reference database onto the taxonomic nomenclature of another via rank-agnostic exact name matching. Figure 2 provides an example to demonstrate the mapping procedure used by *taxmapper* for several different variations in user-specified parameters. For each ASV in a data set, the taxonomic name at the lowest annotated rank (e.g., species if annotated, otherwise genus, etc.) is compared to all taxonomic names at the lowest annotated rank of the target taxonomic nomenclature. If an exact match is found in the reference taxonomic nomenclature onto which assignments are being mapped, the ASV is assigned the matched taxonomic name along with all higher taxonomic names according to the reference nomenclature, is not assigned (assigned NA) at all lower ranks, and mapping is complete for the ASV (Fig. 2). If an exact match is not found, the name is searched at higher ranks within the target taxonomic nomenclature. If an exact match is still not found, depending on user-controlled inputs (see below and Fig. 2) *taxmapper* either: collects taxonomic synonyms for the name being mapped, reformats the name and its taxonomic synonyms, and searches for each possible alternative name until a match is found; or, repeats the search for the taxonomic name assigned to the ASV at the next lowest annotated rank (e.g., genus if the species assignment was not successfully mapped). *taxmapper* returns the input ASV-identifying data with taxonomic assignments mapped onto the specified reference database's taxonomic nomenclature (Fig. 2). If specified by the user, *taxmapper* also returns the taxonomic names for which no exact match was found in the reference taxonomic nomenclature, as well as the "mapping rubric" containing all unique ASV taxonomic assignments supplied by the user and their corresponding mapped taxonomic assignments using the reference nomenclature.

The approach taken by *taxmapper* for mapping taxonomic assignments implicitly assumes that a taxonomic name has equivalent meaning regardless of the reference database in which the taxonomic name is found. This assumption is violated in reference nomenclatures that employ ambiguous terms as standalone taxonomic names such as "incertae sedis", "Clade_X", "Group_2", etc. (a complete list is provided in the package documentation). In the event such taxonomic names are assigned to an ASV and encountered by *taxmapper*, the taxonomic name at the lowest unambiguously annotated rank is appended to the ambiguous name. Exact name matching then proceeds as described above for the newly created, unambiguous taxonomic name (Fig. 2).

Several optional arguments may be supplied to *taxmapper* to increase the number of taxonomic names mapped (Fig. 2). Users may relax the exact name-matching employed by *taxmapper* with the *ignore.format* argument. This relaxation attempts to account for common formatting differences between reference databases (such as the interchangeable use of underscores, hyphens, or spaces, as in *Pseudonitzchia* vs. *Pseudo-nitzchia* in Fig. 2;

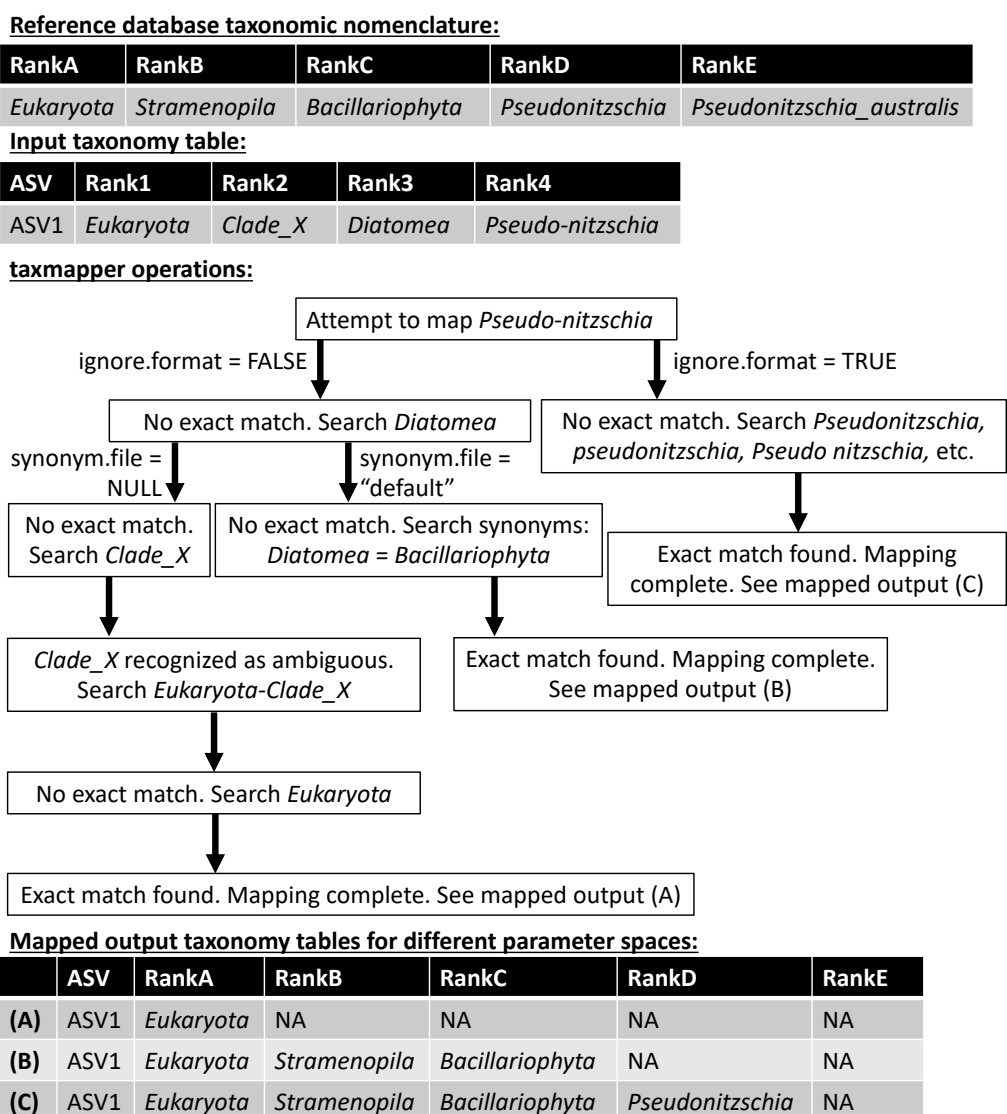

**Reference database taxonomic nomenclature:**

| RankA | RankB | RankC | RankD | RankE |
|---|---|---|---|---|
| *Eukaryota* | *Stramenopila* | *Bacillariophyta* | *Pseudonitzschia* | *Pseudonitzschia_australis* |

**Input taxonomy table:**

| ASV | Rank1 | Rank2 | Rank3 | Rank4 |
|---|---|---|---|---|
| ASV1 | *Eukaryota* | *Clade_X* | *Diatomea* | *Pseudo-nitzschia* |

**taxmapper operations:**

Attempt to map *Pseudo-nitzschia*

ignore.format = FALSE

ignore.format = TRUE

No exact match. Search *Diatomea*

No exact match. Search *Pseudonitzschia, pseudonitzschia, Pseudo nitzschia,* etc.

synonym.file = NULL

synonym.file = "default"

No exact match. Search *Clade_X*

No exact match. Search synonyms: *Diatomea = Bacillariophyta*

Exact match found. Mapping complete. See mapped output (C)

*Clade_X* recognized as ambiguous. Search *Eukaryota-Clade_X*

Exact match found. Mapping complete. See mapped output (B)

No exact match. Search *Eukaryota*

Exact match found. Mapping complete. See mapped output (A)

**Mapped output taxonomy tables for different parameter spaces:**

| | ASV | RankA | RankB | RankC | RankD | RankE |
|---|---|---|---|---|---|---|
| **(A)** | ASV1 | *Eukaryota* | NA | NA | NA | NA |
| **(B)** | ASV1 | *Eukaryota* | *Stramenopila* | *Bacillariophyta* | NA | NA |
| **(C)** | ASV1 | *Eukaryota* | *Stramenopila* | *Bacillariophyta* | *Pseudonitzschia* | NA |

**Figure 2** **Schematic diagram illustrating the approach employed by the taxmapper algorithm to map taxonomic assignments onto different taxonomic nomenclatures.** An example input taxonomic assignment and synonymous (through rankD and rank4) entry in a different taxonomic nomenclature are provided as inputs to the *taxmapper* algorithm. Three examples of mapped taxonomic assignments (A, B, C) are shown based on adjustments to *taxmapper* parameters indicated by arrows in the flow chart.

a complete list of formatting variants for which exact matches are searched is available in the package documentation).

Users may also consider taxonomic synonyms supplied with the ensembleTax package, or provide *taxmapper* with a custom synonym file that includes a collection of taxonomic names and known synonyms. Synonyms are searched in the event that an exact match for a particular taxonomic name is not found in the taxonomic nomenclature onto which the name is being mapped (Fig. 2). The ensembleTax package includes a compilation
of eukaryotic taxonomic synonyms that were compiled manually based on the NCBI Taxonomy Browser (*Benson et al., 2012*), the World Register of Marine Species (*WoRMS Editorial Board, 2020*), Wikispecies (https://species.wikimedia.org/wiki/Main_Page), the Integrated Taxonomic Information System (http://www.itis.gov), the Tree of Life Web Project (*Maddison & Schulz, 2007*), and various other literature sources (*Silén, 1972; Cavalier-Smith, 1993; Adl et al., 2005; Casu & Curini-Galletti, 2006; Gómez, Moreira & López-García, 2010; Ratnasingham & Hebert, 2007; Adl et al., 2012; Braun, 2018; Hibbett et al., 2018; Varol et al., 2018; Adl et al., 2019*). AlgaeBase (*Guiry & Guiry, 2020*) was used to identify some sources of primary literature containing taxonomic synonyms cited above. Users may also supply custom collections of taxonomic synonyms for use with *taxmapper*.

If taxonomic names at the basal rank should be retained in the mapped taxonomy table regardless of whether they are found in the taxonomic nomenclature onto which they are being mapped, these may be supplied to *taxmapper* as exceptions. These names will be assigned to the basal rank of the mapped taxonomy table (kingdom or domain in most reference database taxonomic nomenclatures), with all other ranks unassigned. This option is intended for cases where assignments of non-target organisms are important. For example, if a universal primer set is used to amplify a marker gene from both eukaryotes and prokaryotes (e.g., the 16S and 18S rDNA as in *Parada, Needham & Fuhrman, 2016*), it may be necessary to retain prokaryote assignments when mapping onto a eukaryote-specific taxonomic nomenclature, or vice-versa.

The *assign.ensembleTax* algorithm computes ensemble taxonomic assignments based on any number of input taxonomy tables that share a common taxonomic nomenclature. Figure 3 illustrates several example ensemble determinations based on two different examples of input taxonomic assignments and considering adjustments to several *assign.ensembleTax* parameters. Ensemble assignments are computed independently for each ASV and each taxonomic rank by finding the highest-frequency taxonomic assignment across all input taxonomy tables. If particular taxonomy tables are likely to include more robust taxonomic assignments, the user may weight taxonomic assignments in these tables more highly than others using the *weights* argument. In the event that two or more taxonomic assignments are found at equal weighted (based on the *weights* argument) maximum frequencies, the ensemble taxonomic assignment is unassigned (assigned NA; Fig. 3). This behavior can be modified by specifying the names of taxonomy tables whose assignments should be prioritized in the event that two or more taxonomic assignments are found at equal weighted maximum frequencies (the *tiebreakz* argument; Fig. 3A). By default, non-assignments (represented by NA) are considered when determining ensemble taxonomic assignments, and the ensemble is assigned NA when this is the highest-frequency assignment (Fig. 3). Non-assignments can be excluded from ensemble taxonomic assignment calculations using the *count.na* argument (Fig. 3B). Finally, users may specify the minimum proportion of input taxonomy tables that a taxonomic assignment must be found in in order to be assigned as the ensemble taxonomic assignment using the *assign.threshold* argument (Fig. 3B).

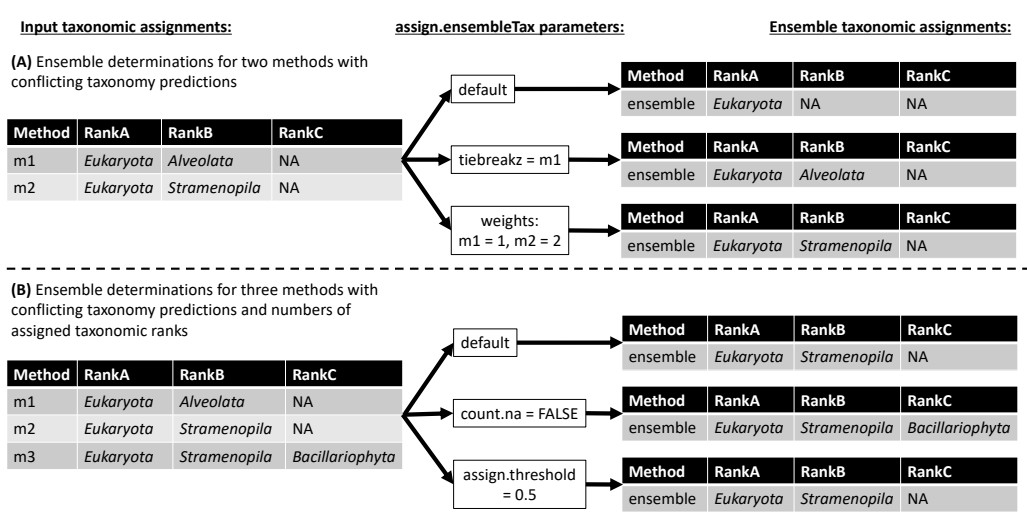

**Figure 3** **Schematic diagram demonstrating ensemble taxonomic assignment determinations by the** ***assign.ensembleTax*** **algorithm for different combinations of input taxonomic assignments and arguments.** (A) Example ensemble taxonomic assignment determinations for a single ASV based on two conflicting methods, illustrating the potential for ensemble methods to remove annotation errors introduced by individual taxonomic assignment methods and/or for users to select which taxonomic assignment methods are prioritized in the event of disagreements between individual methods. (B) Example ensemble taxonomic assignment determinations for a single ASV based on three individual methods that vary in both the number of ranks with assigned names and the identity of names where they are assigned.

# APPLICATION OF ENSEMBLETAX TO NATURAL MARINE PROTIST COMMUNITIES

## Data set overview and application of the ensembleTax package

To demonstrate that the ensembleTax package can be used to compute ensemble taxonomic assignments optimized for a variety of scientific objectives, we used the ensembleTax R package to determine ensemble taxonomic assignments for 15447 protistan ASVs inferred from 358 samples of marine plankton communities from the Santa Barbara Channel, CA. The ASVs are derived from the V9 hypervariable region of the 18S rDNA. Wet lab methods are described in *Catlett et al. (2020)*. ASVs were determined using dada2 (*Callahan et al., 2016*) following *Catlett et al. (2020)*, and four independent taxonomic assignments were determined for each ASV by implementing the RDP naïve Bayesian classifier (*Wang et al., 2007*) with a bootstrap confidence threshold of 60% and the idtaxa algorithm (*Murali, Bhargava & Wright, 2018*) with a bootstrap confidence threshold of 50% against both the pr2 and silva reference databases. These four sets of taxonomic assignments are referred to as bayes-pr2, bayes-silva, idtax-pr2, and idtax-silva. Assignments generated using the silva database were mapped onto the pr2 taxonomic nomenclature, and assignments generated with the pr2 database were mapped onto the silva taxonomic nomenclature using *taxmapper* and considering the taxonomic synonyms included with the ensembleTax package, as well as formatting variants of these names (*ignore.format =TRUE, synonym.file = "default"*). *Bacteria* and *Archaea* assignments were maintained when mapping bayes-silva

and idtax-silva onto pr2 since pr2 does not include these domains/kingdoms (*exceptions =c("Archaea", "Bacteria")*).

In order to illustrate that *assign.ensembleTax* parameters can be adjusted to balance trade-offs between obtaining more ASVs with assigned taxonomy or more robust taxonomic assignments supported by multiple methods, two different collections of ensemble taxonomic assignments were computed from the four independent taxonomy tables generated with each reference databases' taxonomic nomenclature. The first favored ensemble taxonomic assignments supported by multiple individual methods (ensemble1-pr2 and ensemble1-silva), and the second favored obtaining more predicted clade names for a larger proportion of ASVs (ensemble2-pr2 and ensemble2-silva). Annotations supported by multiple independent methods were prioritized in ensemble1-pr2 and ensemble1-silva by including non-assignments in ensemble determinations (*count.na =TRUE*), and by specifying that no single taxonomy table should be favored in the event that different taxonomic assignments were found at equivalent maximum frequencies across the input taxonomy tables (*tiebreakz = NULL*). An increased proportion of ASVs with more predicted clade names was prioritized in ensemble2-pr2 and ensemble2-silva by ignoring non-assignments (*count.na =FALSE*), and by prioritizing assignments from specific taxonomy tables when multiple taxonomic assignments were found at equivalent maximum frequencies across the input taxonomy tables. In such events, assignments found in idtax-pr2, idtax-silva, and bayes-pr2, respectively, were prioritized in ensemble2-pr2 (*tiebreakz =c("idtax-pr2", "idtax-silva", "bayes-pr2")*), while idtax-silva, idtax-pr2, and bayes-silva, respectively, were prioritized in ensemble2-silva (*tiebreakz = c("idtax-silva", "idtax-pr2", "bayes-silva")*). In order to focus our analysis on protists, ASVs assigned to *Archaea*, *Bacteria*, *Metazoa*, *Fungi*, or *Streptophyta* according to ensemble2-pr2, as well as ASVs that were <90 or >180 nt (target amplicon is 120–130 nt) in length, were discarded prior to further analysis.

## Results

We compared the taxonomic assignments obtained with each of the four individual methods (bayes-pr2, bayes-silva, idtax-pr2, idtax-silva) to the two collections of ensemble taxonomic assignments (ensemble1 and ensemble2) using both the pr2 (Figs. 2A, 2C, 2E) and silva (Figs. 2B, 2D, 2F) taxonomic nomenclatures. To investigate differences in the number of predicted clade names between the ensemble and individual taxonomic assignment methods at different levels of the taxonomic hierarchy, we performed rank-wise comparisons of the proportion of ASVs that remained unassigned in each taxonomy table (Figs. 2A–2B). We also calculated the proportion of ASVs for which taxonomic assignments predicted with one individual or ensemble method were assigned to more or less ranks, perfectly agreed, or disagreed (at any rank) with each of the two collections of ensemble taxonomic assignments (Figs. 2C–2F). The latter comparisons were performed to assess the rates of agreement and disagreement between individual and ensemble methods using different *assign.ensembleTax* parameters.

Comparisons of the proportion of unassigned ASVs at each taxonomic rank across the six taxonomy tables demonstrate that substantially more taxonomic names can be

assigned to ASVs at all taxonomic ranks with certain implementations of the ensembleTax package (Figs. 2A and 2B). The ensemble2 taxonomy table had the lowest proportion of ASVs unassigned at all ranks across both the silva and pr2 taxonomic nomenclatures, with the largest increase in the number of taxonomic names assigned at higher ranks (kingdom/domain to class). Notably, >13% more ASVs were assigned to a class in the ensemble2 taxonomy table relative to any other taxonomy table using both taxonomic nomenclatures. With the silva taxonomic nomenclature, nearly 20% more ASVs were assigned to a class in the ensemble2 taxonomy table than with any other method. Conversely, in the ensemble1 taxonomy table the proportion of ASVs that remained unassigned was typically higher than all but one or two of the other taxonomy tables. However, where taxonomic names are assigned in the ensemble1 taxonomy table, they are expected to be robust as they were predicted by two or more of the independent taxonomic assignment methods.

Comparisons of the taxonomic assignments predicted for each ASV by each independent taxonomic assignment method with the two collections of ensemble taxonomic assignments demonstrate that the parameter space of the *assign.ensembleTax* algorithm can be optimized to address different scientific objectives (Figs. 2C–2F). In comparisons with the ensemble1 taxonomy table, a higher proportion of ASVs were assigned taxonomy at more ranks with individual taxonomic assignment methods (9.3–49.1% with the pr2 nomenclature; 22.2–32.8% with the silva nomenclature) than were assigned at less ranks with individual methods (6.3–27.4% with the pr2 nomenclature; 3.1–19.5% with the silva nomenclature). This result reflects the conservative approach used to compute ensemble assignments in the ensemble1 taxonomy table. Conversely, relative to the ensemble2 taxonomy table, 30% or more of the ASVs were assigned taxonomy at less ranks by all independent taxonomic assignment methods, again demonstrating the substantial increase in the number of ASVs that can be assigned taxonomy at lower ranks with less conservative implementations of the ensembleTax package.

Generally, where taxonomic names were assigned, rates of disagreement were low (<10%) for all independent taxonomic assignment methods when compared with either set of ensemble assignments. Rates of disagreement were lower in comparisons with the ensemble1 taxonomy table (0.7−2.2% with the pr2 nomenclature; <2% with the silva nomenclature except for 6.6% for idtax-silva) than in comparisons with the ensemble2 taxonomy table (2.2−5.8% with the pr2 nomenclature; 3.9−8.4% with the silva nomenclature). This result suggests that predicting more clade names for more ASVs with less conservative *assign.ensembleTax* implementations comes at the cost of greater uncertainty in taxonomy predictions and a likely increase in false positive assignments. Interestingly, no conflicting taxonomic assignments were found when comparing the two ensemble taxonomy tables with one another for either taxonomic nomenclature, meaning the two *assign.ensembleTax* parameter spaces resulted in identical taxonomic assignments where ASVs were assigned. Differences in the number of ASVs with taxonomy assigned at lower ranks were large however, with 67 and 70% of ASVs classified to a lower rank in the ensemble2 taxonomy table for the silva and pr2 taxonomic nomenclatures, respectively.

## DISCUSSION

### Optimizing ensemble assignments for different scientific objectives

The ensembleTax R package allows analysts of phylogenetically-informative marker gene sequence data to compute ensemble taxonomic assignments by integrating taxonomic assignments predicted with any number of independent methods (Figs. 1–3). The use of multiple reference databases employing disparate taxonomic nomenclatures is enabled by the *taxmapper* algorithm, which maps one taxonomic nomenclature onto another by rank-agnostic exact name matching (Fig. 2). The *assign.ensembleTax* algorithm computes ensemble taxonomic assignments and includes a suite of parameters that can be modified to optimize trade-offs between predicting more clade names for more ASVs or only assigning taxonomic names that are supported by multiple independent methods (Fig. 3).

Application of the ensembleTax package to determine ensemble taxonomic assignments for a large collection of protistan ASVs sampled from the coastal ocean confirmed that different parameter spaces can be implemented in the ensembleTax package to optimize ensemble assignments for different scientific objectives (Fig. 4). Conservative implementations of *assign.ensembleTax* can be achieved by counting non-assignments (*count.na* =TRUE), by leaving an ASV unassigned when conflicting assignments are found at equal maximum frequencies across the independent assignment methods considered (*tiebreakz* = NULL), and/or by increasing the proportion of independent methods that must predict a taxonomic assignment for it to be assigned to the ensemble (*assign.threshold* = 0.5 or *assign.threshold* =1). Conservative ensemble determinations result in minimal disagreements in taxonomy predictions between the ensemble and independent methods indicating strong support for taxonomy predictions across methods (Figs. 4C–4D), but also result in fewer ASVs with predicted taxonomic names in the ensemble (Fig. 4). These methods are thus well-suited to scientific applications that require more robust, well-supported taxonomy predictions and where obtaining taxonomy predictions for fewer ASVs can be tolerated, such as studies that rely heavily on taxonomic assignments at lower ranks and/or that focus on the diversity or ecology of a particular taxonomic group.

Conversely, less conservative ensemble assignments can be determined by removing non-assignments from consideration (*count.na* =FALSE), by prioritizing certain methods in the event that conflicting assignments are found at the highest frequency (*tiebreakz*, *weights*), and/or by reducing the proportion of taxonomy tables that must corroborate a taxonomic assignment for it to be prescribed to the ensemble (*assign.threshold* =0). Such implementations increase the number of ASVs with assigned taxonomic names at all ranks, but also increase the number of ASVs with conflicting taxonomic assignments between individual methods and the ensemble (Fig. 4). Thus, less conservative implementations of the ensembleTax package are better suited for scientific applications where assigning taxonomy to a higher proportion of ASVs is needed and where false positive annotations can be tolerated, such as studies focused on very broad taxonomic groupings (supergroups, divisions, etc.). However, it should be noted that the increase in ASVs with predicted taxonomy comes at the likely expense of an increase in incorrect taxonomic assignments. Therefore, while taxonomic assignments of ASVs at lower ranks (e.g., family, genus) should

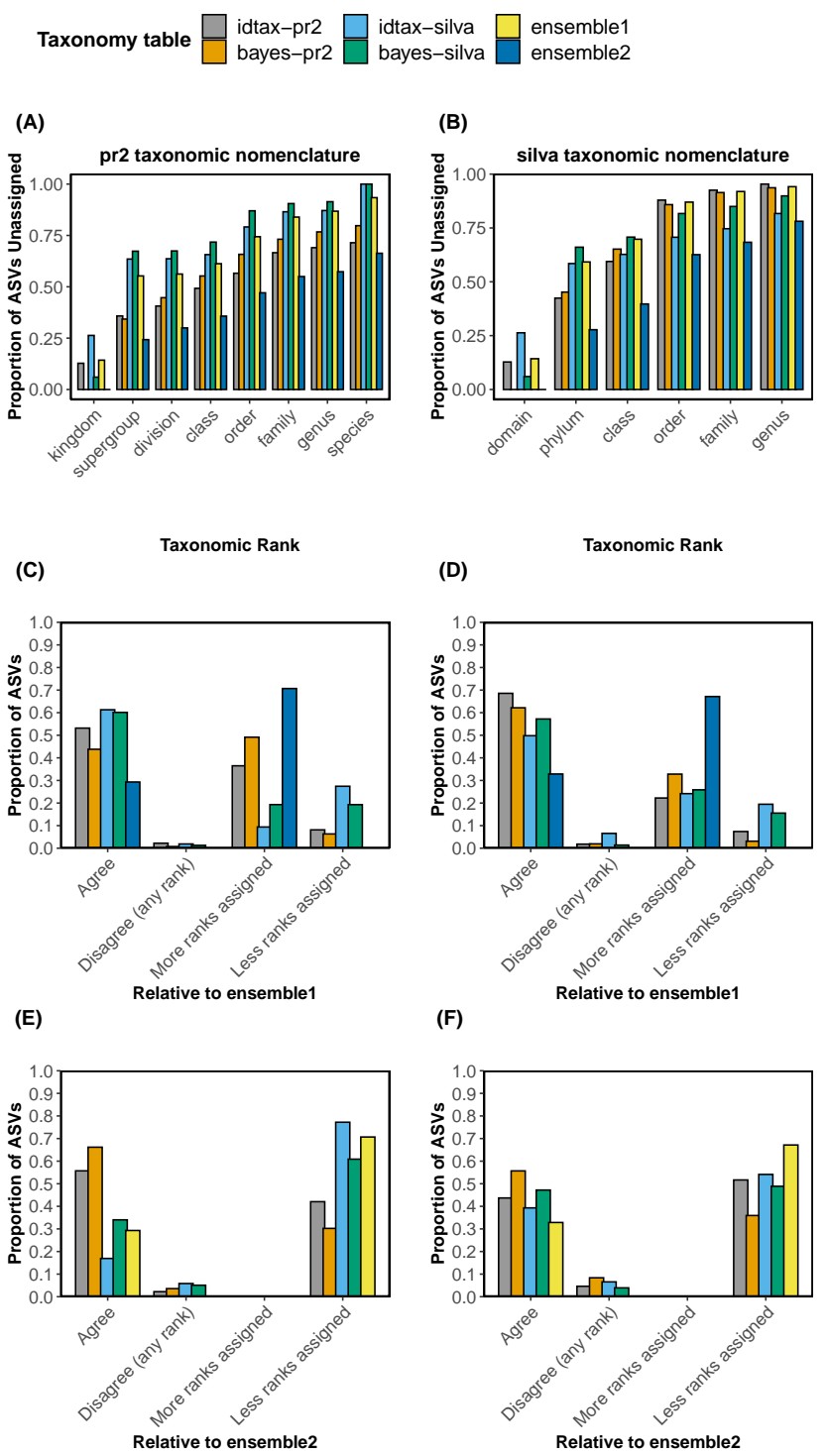

**Figure 4 Comparisons of taxonomic assignments predicted by individual methods with ensemble taxonomic assignments for a large data set of protistan ASVs sampled from the coastal ocean.** (A, B) Rank-wise comparisons of the proportion of ASVs that remained unassigned 

**Figure 4 (…continued)**
(bootstrap values < 60% in bayes-pr2 and bayes-silva, <50% in idtax-pr2 and idtax-silva) in each taxonomy table using (A) the Prostistan Ribosomal Reference Database v4. 12.0 (pr2) and (B) the SILVA SSU nr reference database v138 (silva) taxonomic nomenclatures. (C, D, E, F) Compare the taxonomic assignments predicted by individual (bayes-pr2, bayes-silva, idtax-pr2, idtax-silva) and ensemble (ensemble1, ensemble2) methods with taxonomic assignments predicted by ensemble methods favoring (C, D) more robust taxonomic assignments supported by multiple methods or (E, F) assigning taxonomy to more ranks for a higher proportion of ASVs using the (C, E) pr2 or (D, F) silva taxonomic nomenclatures. In (C, D, E, F), "Agree" indicates that the two taxonomic assignment methods agree in both the number of ranks assigned and the identity of assigned taxonomic names. "More/less ranks assigned" indicates that where taxonomic names are assigned they agree across the two methods, but more or fewer clade names were predicted by one method than the other.

generally be interpreted with caution (*Murali, Bhargava & Wright, 2018*; *Edgar, 2018a*), this is especially true for less conservative implementations of the ensembleTax package. Overall, the ensembleTax package enables computation of optimized ensemble taxonomic assignments for a wide variety of scientific questions.

## Potential for ensemble methods to increase the accuracy of taxonomy predictions

Recent studies have highlighted errors and conflicts in taxonomic annotations of reference sequences included in widely used reference databases, as well as numerous difficulties associated with realistic validation of taxonomic assignment algorithms (*Edgar, 2018a*; *Edgar, 2018b*; *Murali, Bhargava & Wright, 2018*). While the comparisons of taxonomic assignments of protistan ASVs predicted by individual and ensemble methods above demonstrate the versatility of ensemble methods (Fig. 4), we do not demonstrate nor imply that any individual or ensemble method provides more accurate taxonomy predictions than any other method. Future work should thus be devoted to robust validations of ensemble methods applied to particular marker genes. Nonetheless, in some cases ensemble methods may be expected to reduce the impacts of taxonomic assignment errors introduced by individual methods on downstream analyses of amplicon sequencing data. Here we discuss some example scenarios where ensemble methods may be expected to reduce the propagation of taxonomic assignment errors to downstream analyses.

Ensemble taxonomic assignments may be expected to reduce the impacts of errors in reference database annotations on downstream analyses of amplicon data. Sequence annotation errors are known to exist in large reference databases, and conflicts in the taxonomic annotation of a single reference sequence found in multiple databases have been documented (*Edgar, 2018b*). Erroneous and/or conflicting annotations of a reference sequence should result in incorrect taxonomy predictions for any sampled ASV that is derived from or closely related to the incorrect reference sequence (*Edgar, 2018b*). Figure 3A shows that ensemble methods may mitigate the impacts of such errors on downstream analyses. To illustrate this point, assume the two conflicting taxonomic assignments shown in Fig. 3A arise due to conflicting annotations of the same reference sequence found in different reference databases (e.g., assignments by methods 1 and 2 were generated by implementing the same assignment algorithm with two different reference databases). A conservative (including the default) implementation of the *assign.ensembleTax* algorithm

will not assign taxonomy where the individual methods disagree (Fig. 3A). Since at least one of these assignments must be incorrect, ensemble methods effectively remove the erroneous assignment from downstream analyses. If reference sequence annotation errors are less likely in one reference database than another, the *assign.ensembleTax* parameter space can be modified to favor taxonomic assignments using the reference database that is less error-prone (Fig. 3A). If a reference database is more error-prone for some taxonomic groups and less error-prone for others, this strategy can be extended by computing ensemble assignments separately for different subsets of ASVs. ASVs and their corresponding ensemble taxonomic assignments can then be merged back into a single taxonomy table with a common nomenclature with the use of *taxmapper*. Altogether, this example suggests the ensembleTax package may improve taxonomic assignments by enabling streamlined integration of taxonomy predictions based on multiple reference databases.

Ensemble methods may also be expected to reduce the impacts of taxonomic assignment errors introduced by different assignment algorithms. Taxonomic assignment algorithms are known to introduce various types of error in taxonomic assignments of ASVs (*Bokulich et al., 2018*; *Edgar, 2018a*; *Murali, Bhargava & Wright, 2018*). Most widely used taxonomic assignment algorithms differ in the analytical approaches used to predict taxonomy, resulting in differences in observed error rates across a variety of validation exercises (*Bokulich et al., 2018*; *Edgar, 2018a*; *Murali, Bhargava & Wright, 2018*). Ignoring annotation errors in reference databases (see above), it is thus reasonable to assume that error profiles associated with different sequence classification approaches are independent from one another. Independent error profiles across different taxonomic assignment algorithms should lead to situations where a single assignment algorithm predicts an incorrect taxonomic assignment for an ASV that is not corroborated by other assignment algorithms (e.g., the assignment by method "m1" in Fig. 3B may be erroneous). In these situations, conservative implementations of the *assign.ensembleTax* algorithm can again reduce the impacts of taxonomic assignment errors on downstream analyses (Fig. 3B). Overall, the above conceptual examples demonstrate the potential for ensemble methods to improve accuracy in taxonomic assignments of marker gene sequence data, though further research is required to evaluate the accuracy of ensemble methods relative to individual methods and to optimize ensemble methods for specific marker genes and scientific questions.

## CONCLUSIONS

We present the ensembleTax R package, including algorithms for flexible computations of ensemble taxonomic assignments of phylogenetic marker gene sequence data. The two core algorithms, *taxmapper* and *assign.ensembleTax*, compute ensemble taxonomic assignments from any combination of taxonomic assignment algorithms and reference databases (Figs. 1–3). The package data includes pre-compiled taxonomic nomenclatures from several widely used reference databases, as well as a collection of eukaryotic taxonomic synonyms that can improve the performance of *taxmapper* when used with eukaryotic

ASVs. Use of the ensembleTax package to compute ensemble taxonomic assignments on a eukaryotic 18S-V9 rDNA data set from the coastal ocean showed that parameters can be tuned to optimize ensemble taxonomic assignments for specific scientific questions and objectives (Fig. 4). Further development of the ensembleTax package will continue to expand the breadth of taxonomic assignment algorithms and reference databases explicitly supported, and to ensure the most up-to-date versions of reference databases remain easily accessible. Contributions from the community are welcome in these areas. Given the potential of ensemble methods to improve taxonomic assignments of marker gene sequences, future work should be devoted to robust evaluations of the performance of ensemble taxonomic assignment methods relative to that of individual methods, and to determining optimal ensemble inputs and parameters for particular groups of organisms, phylogenetic marker genes, and scientific questions.

## ACKNOWLEDGEMENTS

We acknowledge the efforts of the authors and maintainers of the R packages ensembleTax relies on, as well as the R Core Team. We thank Jasmine Wang for assistance compiling taxonomic synonyms, and Nicholas Baetge, Logan Kozal, Dave Siegel, Debora Iglesias-Rodriguez, and Lizzy Wilbanks for helpful conversations and their general encouragement of the pursuit of this work. Finally, we thank the Plumes and Blooms team for assistance collecting the environmental samples analyzed here, and Paul Matson and Tom Lankiewicz for assistance with sample processing.

### Funding

This work was supported by the National Aeronautics and Space Administration Biodiversity and Ecological Forecasting program (Grant NNX14AR62A), the Bureau of Ocean and Energy Management Ecosystem Studies program (BOEM award MC15AC00006), and NOAA in support of the Santa Barbara Channel Biodiversity Observation Network. Additional support for Dylan Catlett was provided by a NASA Earth and Space Science Fellowship (Grant NNX16AO44HS02), and additional support for Connie Liang was provided by the UC Santa Barbara Coastal Fund (Grant SPR19-18). The funders had no role in study design, data collection and analysis, decision to publish, or preparation of the manuscript.

### Grant Disclosures

The following grant information was disclosed by the authors:
National Aeronautics and Space Administration Biodiversity and Ecological Forecasting program: NNX14AR62A.
Bureau of Ocean and Energy Management Ecosystem Studies Program: MC15AC00006.
NOAA.
NASA Earth and Space Science Fellowship: NNX16AO44HS02.
UC Santa Barbara Coastal Fund: SPR19-18.
## Competing Interests

The authors declare there are no competing interests.

## Author Contributions

- Dylan Catlett conceived and designed the experiments, performed the experiments, analyzed the data, prepared figures and/or tables, authored or reviewed drafts of the paper, and approved the final draft.
- Kevin Son and Connie Liang conceived and designed the experiments, performed the experiments, analyzed the data, authored or reviewed drafts of the paper, and approved the final draft.

## Data Availability

The raw data, taxonomic assignments output, and the R code used to produce Fig. 4 are available at GitHub: https://github.com/dcat4/ensembleTax_analysis.

The raw sequence data are also available at NCBI: PRJNA532583.

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
