# Peer review of "ensembleTax: an R package for determinations of ensemble taxonomic assignments of phylogenetically-informative marker gene sequences"

_PeerJ, doi:10.7717/peerj.11865_

## Round 0.1 · original submission · Major Revisions

Dear Dr. Catlett and colleagues:

Thanks for submitting your manuscript to PeerJ. I have now received three independent reviews of your work, and as you will see, the reviewers raised some concerns about the research. Despite this, these reviewers are optimistic about your work and the potential impact it will have on research studying automated approaches to taxonomic assignments for genes and proteins. Thus, I encourage you to revise your manuscript, accordingly, taking into account all of the concerns raised by both reviewers.

Your manuscript needs to be restructured and streamlined to improve clarity and presentation. The reviewers collectively provide many helpful tips. Also, please ensure that all claims are backed up by evidence (either your own or from the literature). Missing references deemed important by the reviewers should be included.

Please note that reviewer 3 has included a marked-up version of your manuscript.

There are many suggestions, which I am sure will greatly improve your manuscript once addressed.

I look forward to seeing your revision, and thanks again for submitting your work to PeerJ.

Good luck with your revision,

-joe

·

Basic reporting

Writing is hard to follow. Some essential background is missing.

Experimental design

I was not able to understand the goals of the algorithms or the details of how the algorithms work.

Validity of the findings

Some implicit claims are not adequately supported by results.

Additional comments

Abstract
The term "resolution" should not be used here. It implies improved distinction between categories, overlooking the fact that most microbes have not been named by taxonomists, so the default assumption should be that more names would mostly be false positives. Thus, tuning to predict more names is quite likely a bad idea, while it naively sounds good. "Number of predicted clade names" is more understandable, but may still be misleading for readers who are unaware that a missing name is often the correct prediction.
"Fidelity" is an infelicitous term in this context. It is not helpful to define accuracy=fidelity without further explanation.
The claim that users can prioritize "accuracy", however defined, is not supported by the results in the paper. Defining and measuring accuracy of taxonomy prediction from sequence is ferociously difficult and controversial, as shown by several cited papers including a couple of mine.
A more supportable statement would be something like "Thresholds can be set by the user to choose between obtaining more predicted clade names at lower ranks, some of which may be false positives, versus fewer predictions where missing predictions may be false negatives."
However, this trade-off can already be made by using one good algorithm, e.g. setting the bootstrap cutoff with the RDP Classifier, which begs the question: what then is the motivation or benefit of using the ensemble classifier described by the paper? There are many examples in the machine learning literature where using meta-classifiers or ensemble classifiers achieve higher accuracy, but simply implementing an ensemble classifier does not guarantee that better results will be achieved.
The design and reporting of the validation on protist data is problematic. The description reads as a series of software commands without adequate motivation and interpretation. Metrics are not clearly defined, e.g. the crucial concept of a "misclassification" is defined only in a figure caption. It is misleading to define a misclassification as a disagreement with the consensus, because the prediction could be correct and the consensus is wrong, or both the prediction and the consensus could be wrong.
Better would be to start with a discussion of the design of the validation, which metrics are used and why, and a discussion of how the results should be interpreted. However, this probably cannot be done in a very informative way because the correct taxonomy is not known, leaving us with the ferociously difficult and controversial problem of measuring taxonomy prediction accuracy from sequence. Presented as an illustration of how the method can be used in practice, noting conflicts between different classifiers as a plausibility argument for taking a consensus would be fine, but the text should be written carefully to avoid over-stating claims about advantages of this approach which are not supported by the results.
Given the opportunity for misunderstanding, the Abstract and main text should make clear that there is no evidence that the described ensemble classifier is more accurate by any metric compared to using one algorithm such as the RDP Classifier.
I had some difficulties following what exactly the package does and why it does it, so I may have missed some salient points. With that said, it seems to me that the ensemble classifier described in the paper adds a layer of complication for no benefit to the user. If so, perhaps the paper could be improved by focusing on nomenclature reconciliation.
There are many possible approaches to combining databases with different sequences, different nomenclatures and different annotation file formats. One is to convert annotations into the format required by particular software packages (say, the RDP Classifier) then run a package on the same set of query sequences with all databases, and as a final step attempt to reconcile the predictions. Another is to reconcile the databases into one set of sequences with a harmonized set of annotations. Then a package such as the RDP Classifier can be run directly on the combined database. Some consideration of these alternatives and motivations for the authors' choices would be useful.
As best I understand it, the authors take the first approach, with the additional feature that multiple classification algorithms can be combined (I suspect this is overcomplicating the solution). I suggest that the second approach would be a more valuable contribution, because it would be plug-and-play in a wide variety of pipelines. I'm a good example -- I'm a command-line / C++ hacker and occasional Python coder with a strong aversion to R because I find it deeply obscure, so I would need a very strong reason to consider using an R package. By contrast, combined and reconciled databases for popular markers such as 16S, 18S and ITS with a choice of annotations in widely-used formats (SILVA, usearch, RdRP Classifier...) would be plug-and-play replacements that could be instantly adopted by users of many different software environments. I would happily recommend such databases to usearch users.
[82] internal transcribed spacer gene (ITS)
ITS is not a gene, it is a transcribed but non-coding region between genes, similar to an intron.
[85] inferred operational taxonomic units (OTUs) or amplicon sequence variants (ASVs)
An OTU as such cannot be "inferred" as it is an artificial construct. An ASV is a special case of an OTU construction where the sequences are clustered at 100% identity, not a separate category.
[86] as representatives of individual microbial species within a community
An OTU or ASV may represent multiple species having the same sequence, this is common with shorter ASVs such as 16S V4.
[88] Because an organism’s taxonomy is often linked to its ecological niche within an ecosystem, assigning taxonomic identities to inferred OTU/ASV sequences imparts ecological significance to genetic data. Taxonomic assignment thus represents a critical component of all marker gene sequencing studies.
This is doubtful in the case of most prokaryotes and fungi. Consider for example E. coli -- two strains of the same species may share only 60% of their genes with radical consequences for phenotype, e.g. pathogen vs. benign or symbiote. This distinction cannot be made on the basis of a typical 16S amplicon, or even with the full-length 16S gene in many cases. Since taxonomy can often be predicted only to around genus or family rank, I doubt that much can be inferred about ecological role from marker gene taxonomy prediction. This should be re-written, or citations provided to support the claim.
Also, most microbes in databases such as SILVA and UNITE are known only from marker gene sequencing, for the large majority there are no direct observations of phenotype other than a sequence. This issue should be noted, and also that it exacerbates the limitations of inferring anything reliable about phenotype or ecological role by comparing amplicons with sequences in a reference database. Further, most microbial species (genera...) have not been named, which limits the ability of taxonomy prediction to illuminate characteristics of microbes in the sample. This should also be mentioned.
[97] However, different representatives of a single marker gene are often found in different reference databases
This sentence confused me for a while. I think it is trying to say that if two databases for a given gene are compared, they will have some sequences in common and others that are found only in one database. Re-word for clarity.
[102] Meanwhile, a large collection of 18S rDNA reference sequences, some of which are not included in pr2, is also included alongside 16S rDNA reference sequences
This seems irrelevant. The fact that SILVA has both 16S and 18S does not create practical problems for the user as far as I can see. If you have 18S data, use the SILVA 18S reference. Or, use both -- maybe your primers targeting 18S weakly amplified some 16S as well.
[104] Unfortunately, the disparate naming and ranking conventions employed by silva and pr2 force analysts of 18S rDNA data sets to use either pr2 or silva exclusively for their taxonomic assignments.
Not if the user has some coding skills. The goal is for the authors' package to make it easier for the user to combine multiple databases.
[132] ...flexibly compute ensemble taxonomic assignments
While the term "assignment" is commonly used for computational taxonomy predictions from sequence alone, I suggest that "prediction" is better to distinguish a tentative prediction by an error-prone algorithm from expert examination of phenotype observed in vivo or in vitro.
[186] RDP 16 train set
Should be "16S rRNA training set", though "training set" is their jargon and simply "reference" might be better.
[187] Additional assignment algorithms and/or reference databases can be incorporated so long as they are formatted according to the documentation provided with the ensembleTax R package.
Should say more about what "formatted according to the documentation" entails here. Isn't incompatible formatting exactly the problem the ensembleTax package is intended to solve? If the user has the skills for this, why would they use ensembleTax -- they could e.g. run the RDP Classifier in the data2 package.
[135] databases'
Misplaced apostrophe, should be database's.
[Table 1] Taxonomy. An organisms taxonomic affiliation
Missing apostrophe, should be organism's. This definition introduces a new undefined term (how is affiliation different from assignment?) and is regrettably circular.
[203-218]
I was completely baffled by the description of the taxmapper algorithm. I was not able to deduce what problem(s) it is designed to solve -- does it try to reconcile (a) nomenclature only without looking at the sequences, or (b) nomenclature+sequence?
This section should be re-written to give more background. Describe the problems to be solved in reconciling taxonomic nomenclatures. Given illustrative examples, and explain how your approach solves each case. If the method does not consider sequence, then this should be mentioned because sequences introduce new types of conflict between databases which are not apparent from nomenclature alone.
The term "ASV" should not be applied to reference databases, because they were not necessarily obtained by amplicon sequencing (e.g., could be obtained using an HMM to find the gene in a full-length genome assembly).
Rank names are capitalized (Genus, Species...); should be lower case.
If the goal is to reconcile nomenclature+sequence annotations, then presumably, the first step is to identify reference sequences in common between two databases. How is this done, exactly -- presumably by alignment, but what is the method? What happens if one sequence is a substring of the other (i.e., the match has 100% identity but is not full length)? What happens if the best match is close but not exact, say 99%, or the best match is much lower, say 80% id?
[206] What is "lowest possible rank"? There is the lowest rank generally in a given database (e.g., species for SILVA or genus for the RDP training set), but this is not necessarily the same as the lowest rank annotated for a given reference sequence, which might be say family because genus and species are not named.
Note that the interpretation of a blank name is usually not clear -- is this is a positive prediction by the database that the sequence belongs to a novel clade, or was it left blank because there were two alternative named clades and the confidence was too low to distinguish? Databases such as SILVA and Greengenes do not tell us one way or the other.
What about placeholders such as "sp." for an unnamed species, e.g. if two sequences are annotated as "Chroococcidiopsis sp." they could be different species in the same genus. What about "incertae sedis", square brackets to indicate unofficial names, and so on.
Does the method identify conflicting annotations of the same sequence? Unambiguous conflicts do occur in practice (see "Conflicting annotations of the same sequence" in Edgar2018b).
Does the method identify hierarchy conflicts where a clade at a lower rank appears under different parent ranks (e.g., a given genus appears in different families)? If so, how are they resolved? These also happen in practice (see "Nomenclature hierarchy conflicts" in Edgar2018b).
I was equally baffled by the description of the ensembleTax algorithm. In line [248], what is a "taxonomy table"? I suspect this is a table of predicted frequencies for named clades in one sample, but this is not clear from the text. Frequencies cannot be predicted from amplicon sequences because of strong amplification biases such as multiple operons, primer mismatches and so on; see e.g. my paper https://doi.org/10.1101/124149. Would "consensus" classifier be better than "ensemble"? As with other sections in the paper, this reads as a stream of disconnected facts which assumes the reader is already familiar with the problems to be solved and the authors' jargon. It needs better structure and more careful definition of terms and concepts. Lines [280-288] are a good example of a paragraph that is hard to follow with much distracting detail and too little background.
[262]
The terms "under-classification" and "over-classification" should be explained.
[171]
"Data pre-processing" should be explained in more detail.
[183]
abbreviation "nr" should be explained.
[183]
Why were these reference databases chosen? For 16S, using SILVA and Greengenes seems unfortunate because they are mostly based on predicted rather than authoritative taxonomy, and as a consequence have very high error rates as I showed in Edgar2018b. Using the 97% clustered version of Greengenes makes the problem worse because clustering discards many named isolate strains, which are the only sequences with authoritative classifications. Predicting taxonomy from a reference set annotated by unreliable methods and clustered at 97% is asking for trouble in my opinion. Edgar2018b recommends using named isolate strains only as a reference. The authors may disagree, but at a minimum these issues should be raised for the benefit of readers who may not be familiar with these complications.

Reviewer 2 ·

Basic reporting

The text is in general clear but as pointed out in 4 some areas near clarification. Figures need to be improved (see 4).

Experimental design

No comment

Validity of the findings

No comment

Additional comments

# General comments
This paper presents a new R package aimed at reconciling taxonomic assignments of short sequences (metabarcodes) made using different algorithms (dada2 Bayesian assigner or DECIPHER TaxID) or different reference sequence databases (PR2, Silva, RDP) and computing an "ensemble" taxonomy. This package should be useful as metabarcoding is becoming the method of choice for characterizing microbial communities both eukaryotic and prokaryotic. The package has been published on CRAN which means that it has passed software tests successfully. Unfortunately, I did not have the time to test it as extensively as I possibly would have liked. However I think that the package documentation and the paper should undergo a thorough revision in order to improve the package usability and the analyses provided in the paper itself.

* I found the on-line documentation quite minimal and a bit hard to follow. I would suggest to show in a vignette the whole process starting from a small metabarcode sequence dataset and providing the code for the assignement of sequences with dada2 and DECIPHER and then showing how to use the emsembleTax package and interpret the data. I would also suggest that the vignette include plots comparing the output of the different assignments and of the "ensembleTax".
* The description of how the emsembleTax algorithm works should be explained in a separate vignette with examples.
* The description of the function "taxmapper" is not totally clear. Several examples should be given in a vignette. The sentence l. 206 to 208 should be clarified.
* I am not sure that the term "ensemble" taxonomy is really the most appropriate. "ensemble" is in the name of the package which cannot be changed I guess but I would replace this term in the paper itself. I would it rather call "consensus" taxonomy.
* The description of the ensembleTax function algorithm is a bit hard to follow and I would have liked to have an example (maybe in a vignette). What is the "highest-frequency taxonomic assignement" (l. 252) ? If there are only 2 tables, the each table counts for 1 (I understand that you can use the weights argument) ?
* The authors used a V9 dataset to demonstrate the use of the package and to show how it helps improving taxonomic assignement. However I do not think that using V9 is best to demonstrate this as reference sequence databases such as PR2 have much less sequences covering the V9 region than the V4 region. So it would be interesting to complement the analysis with a V4 dataset from the literature (e.g. OSD, Ocean Sampling Day) as well as a bacterial dataset.
* The analysis provided in Fig. 2 is very succinct (see also my comments on the readability of the figure). I would like to understand better how the package improves really assignement, maybe by looking at specific taxonomic groups. Also at least for PR2, the confidence values from DECIPHER are in general lower than the bootstrap values for dada2 assignements. Therefore, this will contribute to the fact that less sequences are assigned with DECIPHER than with dada2 for a given bootstrap level. This may depend though on the way DECIPHER is trained (number of iterations and number of sequences per group)
* The term "misclassification" (l. 319) is a bit misleading. For some taxonomic groups, one database maybe better than the other one, while this is the reverse for another group. None of the function parameters accounts for this, so rather than misclassified maybe use something "disagree with consensus"...

# Specific comments

* l. 239. Algaebase is protected by very strong copyright rules. Does the authors secured an agreement from Algaebase to use the data ?
* Fig. 1 could be made clearer by adopting a more vertical structure...
* Fig. 2 is hard to read. Make bars wider. Move the legend on the top so that the figure can be wider. Explain in the legend what you mean by unassigned, is it anything that has a bootstrap value < 100 ? For panels C to F, it would be better to do cumulative bars for each of the taxonomy table since a given sequence falls into a single of the categories represented in the X axis. Moreover as the panels C to F is shown now the order of the bars is different from the order of the legends... which makes things hard to visualize for color blind people.

Reviewer 3 ·

Basic reporting

The manuscript "ensembleTax: An R package for determinations of ensemble taxonomic assignments of phylogenetically-informative marker gene sequences" is clearly written and provides the necessary references to support their statements and provide context. It is true that maybe the authors should make an effort to reduce the number of references as in some parts it is not necessary to provide multiple references to support a simple statement for example. The figures are correct, however, in figure two the order of the bar plots across the panels, and the legend are not always the same making it more difficult to follow.

Experimental design

It is not clear to me how the concepts of misclassification errors, overclassification errors, and underclassification errors have been defined. From my point of view, they can be only defined by using mock communities or by comparing against a tree built with the classified sequences and based on the tree annotated the taxonomic strings of the tree manually. So, the automatic annotation can be compared against the known mock community or a reliable curated tree, and consequently, the different errors in annotation can be reported. Otherwise, if their definition of the errors is based on the same automatic annotation I do not think they can be called errors, as the authors have not defined what would be the right classification.

Validity of the findings

Their initial premise is in general correct, there are some statements that I do not believe are true, so I would suggest modifying those. The proposed approach and the associated tools are very interesting and I think that part of it can benefit a lot the taxonomic identification of amplicon sequencing data. I find particularly useful the essembleTax part of the essembleTax package (I would like to mention that naming a part of the package as the whole package is somehow confusing, a different name would be helpful). Combining the results of different taxonomic annotation algorithms can be very helpful and I can see how I would use it for my own research. However, I am not so convinced about the taxmapper part. The implicit assumption for taxmapper is that all the reference databases present essentially the same taxonomy and the only real variation lies in the number of ranks. I have a lot of experience parsing and curating reference databases and comparing them and sadly that is not the case. The taxonomies differ, some are curated, some automatized, and when you compared them they can be very different, especially at the family, order, and class level. For certain groups these differences are substantial. So, if I understood how taxmapper works I believe that these divergences in the taxonomies will lead to errors in the annotation. Could the author clarify this (especially if I am wrong because I am not fully understanding what taxmapper is doing)?

Additional comments

'no comment'

Annotated reviews are not available for download in order to protect the identity of reviewers who chose to remain anonymous.

---

## Round 0.2 · accepted · Accept

Dear Dr. Catlett and colleagues:

Thanks for revising your manuscript based on the concerns raised by the reviewers. I now believe that your manuscript is suitable for publication. Congratulations! I look forward to seeing this work in print, and I anticipate it being an important resource for groups studying automated approaches to taxonomic assignments for genes and proteins. Thanks again for choosing PeerJ to publish such important work.

Best,

-joe

·

Basic reporting

-

Experimental design

-

Validity of the findings

-

Additional comments

I'm glad to see the manuscript is much improved. My comments have been satisfactorily address. Couple of errors I notice: line 186 "16S rRNA train set" still not corrected; line 297 "Callahah" should be "Callahan".

Reviewer 2 ·

Basic reporting

OK

Experimental design

OK

Validity of the findings

OK

Additional comments

I did not have the time really to do a new full review of the paper as it has been substantially changed. I think the authors tried in earnest to address all comments made by the 3 reviewers and I think the paper can now be published. It will be complementary of the package and will be useful to its user.